# Resegmentation is an ancestral feature of the gnathostome vertebral skeleton

Katharine E Criswell[1,2]*, J Andrew Gillis[1,2]

[1]Department of Zoology, University of Cambridge, Cambridge, United Kingdom;
[2]Marine Biological Laboratory, Woods Hole, United States

**Abstract** The vertebral skeleton is a defining feature of vertebrate animals. However, the mode of vertebral segmentation varies considerably between major lineages. In tetrapods, adjacent somite halves recombine to form a single vertebra through the process of 'resegmentation'. In teleost fishes, there is considerable mixing between cells of the anterior and posterior somite halves, without clear resegmentation. To determine whether resegmentation is a tetrapod novelty, or an ancestral feature of jawed vertebrates, we tested the relationship between somites and vertebrae in a cartilaginous fish, the skate (*Leucoraja erinacea*). Using cell lineage tracing, we show that skate trunk vertebrae arise through tetrapod-like resegmentation, with anterior and posterior halves of each vertebra deriving from adjacent somites. We further show that tail vertebrae also arise through resegmentation, though with a duplication of the number of vertebrae per body segment. These findings resolve axial resegmentation as an ancestral feature of the jawed vertebrate body plan.

*For correspondence:
kc518@cam.ac.uk

Competing interests: The authors declare that no competing interests exist.

## Introduction

Axial segmentation is key to the body plan organization of many metazoan groups and has arisen repeatedly throughout animal evolution (*Davis and Patel, 1999*). Within vertebrates, the axial skeleton is segmented into repeating vertebral units that provide structural support and protection for soft tissues. Vertebral segmentation is preceded in the embryo by the segmentation of paraxial mesoderm into epithelial blocks called somites (*Figure 1a*). Cells from the ventromedial portion of each somite then undergo an epithelial to mesenchymal transition and migrate around the notochord and neural tube, where they condense into vertebrae.

Cell lineage tracing studies in tetrapods have shown that there is not a 1:1 correspondence between somites and vertebrae. Rather, cells from adjacent somite halves recombine to give rise to a single vertebra, through a process known as 'resegmentation' (*Remak, 1855*). Somite lineage tracing in chick using chick-quail chimeras or lipophilic dyes (*Aoyama and Asamoto, 2000*; *Bagnall et al., 1988*; *Huang et al., 2000*; *Schrägle et al., 2004*; *Stern and Keynes, 1987*; *Ward et al., 2017*) has shown that cells from the rostral half of one somite combine with cells from the caudal half of the adjacent somite to give rise to a single vertebra, with sharp compartment boundaries in the middle of vertebrae reflecting original somite boundaries. Additionally, lineage tracing experiments in axolotl using injections of fluorescent dextrans or grafts of GFP+ somites into wild type hosts point to conservation of resegmentation during vertebral development in lissamphibians (*Piekarski and Olsson, 2014*). However, in teleost fishes, resegmentation is less apparent, with cells from adjacent somite halves undergoing substantial mixing, resulting in vertebrae without clear lineage-restricted compartments (*Morin-Kensicki et al., 2002*). This divergence in the relationship between somites and vertebrae raises the question of whether resegmentation is restricted to tetrapods, or whether it is an ancestral feature of the jawed vertebrate (gnathostome) backbone that has been reduced or lost in teleosts. To answer this question, data are needed from an outgroup to the bony fishes – for example the cartilaginous fishes.

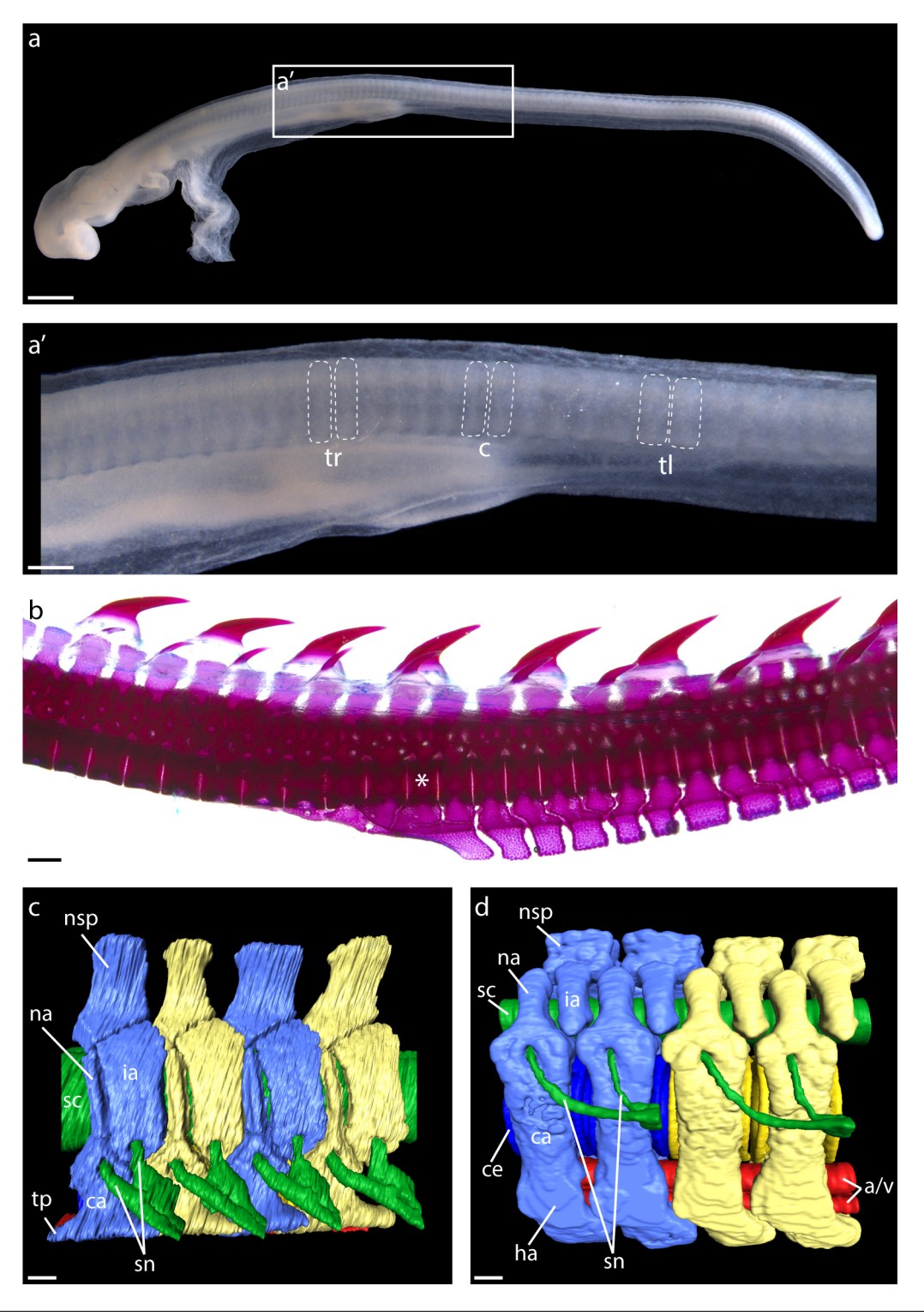

**Figure 1.** Segmental organization of paraxial mesoderm and the vertebral skeleton in the skate. (**a**) Somites of a S24 skate embryo showing (**a'**) trunk (tr), cloacal (c), and tail (tl) somites, delineated with dashed lines. (**b**) A cleared and stained vertebral column from a skate hatchling, with the trunk to tail transition denoted with an asterisk. (**c**) μCT scan reconstruction of skate trunk and (**d**) tail vertebrae, with vertebral elements corresponding to a single body segment color-coded blue or yellow, neural tissue colored green, and blood vessels colored red. Note that in (**d**), one set of spinal nerves spans two vertebrae. a/v, caudal artery and vein; ca, outer cartilage of vertebral

*Figure 1 continued on next page*

*Figure 1 continued*

body; ce, centrum; ha, haemal arch; ia, intercalary arch; na, neural arch; nsp, neural spine; sc, spinal cord; sn, set of spinal nerves; tp, transverse process. Scale bars: (a) 400 µm; (a') 100 µm; (b) 500 µm; (c) and (d) 100 µm.

The vertebral skeleton in cartilaginous fishes consists of a series of neural and intercalary arches and neural spines, tri-layered centra and haemal arches (the latter restricted to the caudal region – *Figure 1b*; *Criswell et al., 2017a*). Notably, the axial skeleton of the embryonic skate forms initially as a continuous, sclerotome-derived cartilaginous tube, which subsequently subdivides into discrete vertebrae (*Criswell et al., 2017a*; *Criswell et al., 2017b*). Elasmobranch cartilaginous fishes (sharks, skates and rays) also show a unique pattern of spinal nerve segmentation along the body axis, with each trunk vertebra containing foramina for both dorsal and ventral spinal nerve roots, while each tail vertebra contains a foramen for only a single nerve root (*Ridewood, 1899*; *Figure 1c,d*). This pattern is widely hypothesized to reflect a complete vertebral duplication in each tail body segment, termed 'diplospondyly'. However, there are currently no experimental data testing this, or, more generally, the relationship between somites and vertebrae, in any cartilaginous fish.

Here, we test whether tetrapod-like resegmentation is the ancestral mode of vertebral segmentation in gnathostomes using an outgroup to the bony fishes, a cartilaginous fish (the little skate, *Leucoraja erinacea*). We also test whether the diplospondyly observed in the tails of cartilaginous fishes arises through duplication of skeletal derivatives from tail vs. trunk somites. We find that skate tail somites do, indeed, duplicate their vertebral derivatives relative to trunk somites, and that resegmentation occurs along the entire body axis. These findings point to a stem-gnathostome origin of axial column resegmentation, and allow us to reconstruct the evolutionary history of vertebral column development.

## Results

### Conservation of rostrocaudal somite polarity in tetrapods and skate

Tetrapod somites exhibit distinct rostrocaudal polarity, and correct rostral and caudal transcriptional identity within a somite is essential for proper resegmentation (*Bussen et al., 2004*; *Hughes et al., 2009*; *Keynes, 2018*; *Leitges et al., 2000*; *Mansouri et al., 2000*). To test for rostrocaudal polarity of somites in skate, we examined expression patterns of *Tbx18* and *Uncx4.1* – tetrapod markers of rostral and caudal somite identity, respectively (*Haenig and Kispert, 2004*; *Kraus et al., 2001*; *Schrägle et al., 2004*) – using wholemount mRNA in situ hybridization. We found that *Tbx18* (*Figure 2a*) and *Uncx4.1* (*Figure 2b*) are both expressed in skate somites at S22, with expression of the former localizing to the rostral somite, and expression of the latter localizing to the caudal somite. To further test polarity of these expression patterns within the somite, we characterized transcript distribution on paraffin sections of somites using multiplexed fluorescent mRNA in situ hybridization, and we found that *Tbx18* and *Uncx4.1* transcripts localize to the rostromedial and caudomedial cells of the somite, respectively (*Figure 2c*). These findings point to a shared molecular basis of somite rostrocaudal polarity between tetrapods and cartilaginous fishes.

### Skates exhibit tetrapod-like resegmentation

To test the relationship between somites and vertebrae in skate, we performed a series of somite fate mapping experiments. We microinjected the ventral portions of two neighboring trunk somites in S24 little skate embryos with either CM-DiI (a red fluorescent analogue of DiI) or SpDiOC$_{18}$ (a green fluorescent analogue of DiO) (*Figure 3a*), and then mapped the contributions of these labeled somites to the vertebral skeleton 8–12 weeks post-injection. We recovered dye in the vertebral cartilages of 27 embryos, with 21/27 embryos retaining both CM-DiI and SpDiOC$_{18}$ labelling in adjacent vertebrae, and 6/27 retaining only CM-DiI. In 22/27 label-retaining embryos, we observed patterns of retention that were consistent with somite resegmentation. For example, in embryos in which both CM-DiI and SpDiOC$_{18}$ were retained, two neighboring trunk somites contributed to a combined total of three vertebrae, with each somite contributing to two vertebral halves (the caudal half of one vertebra and the rostral half of the neighboring vertebra - *Figure 3b–d*). There was little overlap in CM-DiI and SpDiOC$_{18}$ distributions, indicating that cells from adjacent somites undergo little

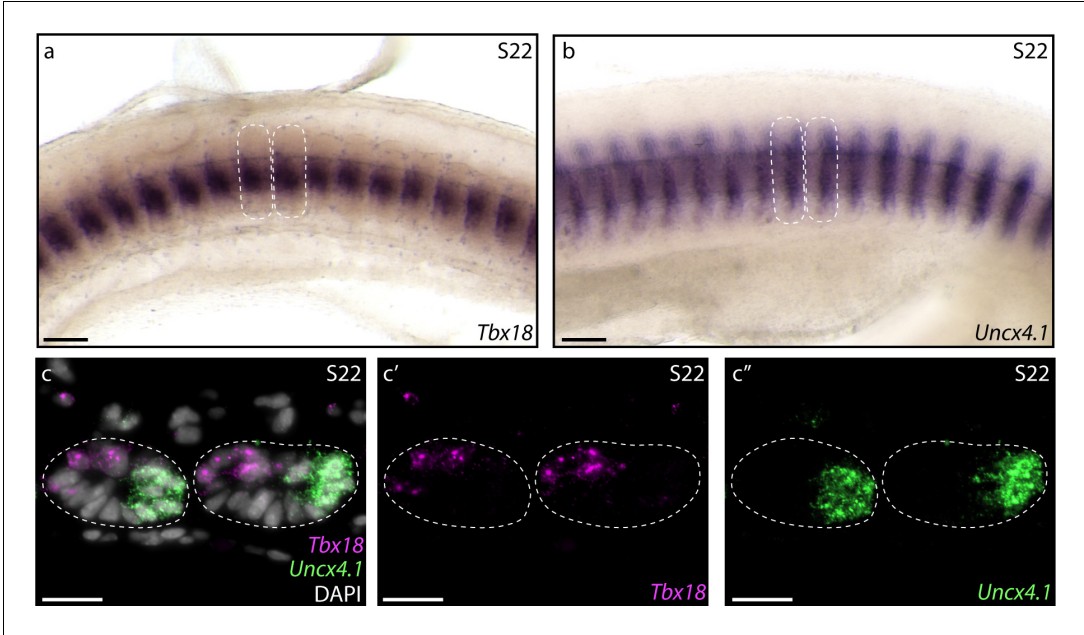

**Figure 2.** Molecular basis of skate somite rostrocaudal polarity. Whole-mount mRNA in situ hybridization reveals (a) rostral expression of *Tbx18* and (b) caudal expression of *Uncx4.1* within skate somites at S22. (c) Hybridization chain reaction (HCR) in situ hybridization on paraffin sections reveals complimentary rostromedial and caudomedial expression of *Tbx18* (magenta, **c'**) and *Uncx4.1* (green, **c"**), respectively, in skate somites at S22. Dashed lines indicate somite boundaries. Scale bars: (a) and (b) 100 μm; (c-c") 25 μm.

mixing, and that somite boundaries are maintained through sclerotome migration, condensation and vertebral differentiation. The results of these labeling experiments are consistent with a tetrapod-like mechanism of resegmentation during the development of trunk vertebrae in the skate.

## Skate tail somites give rise to twice as many vertebrae as trunk somites, and still undergo resegmentation

Elasmobranch fishes show a diplospondylous condition in their caudal vertebrae, in which two vertebrae are present for each set of spinal nerve roots. As each tail vertebra is comparable in anatomical organization to trunk vertebrae (e.g. with neural and intercalary arches – *Ridewood, 1899*), it is speculated that this condition is a consequence of tail somites giving rise to twice as many vertebral units when compared with trunk somites. If somites give rise to double the number of vertebrae in the tail as in the trunk, we would expect approximately equal numbers of trunk somites and vertebrae, while the number of tail vertebrae should far exceed the number of tail somites. We counted trunk and tail somites in S25 skate embryos (at which point somitogenesis has ceased, but somites are still clearly discernable), and compared this with numbers of trunk and tail vertebrae in skate hatchlings. We found that S25 skate embryos possess a mean of 48 trunk somites and 88 tail somites (with the cloaca marking the transition from trunk to tail; n = 17 embryos counted; *Supplementary file 1*), while hatchling skates possess a mean of 48 trunk vertebrae (including the series of fused vertebrae that make up the synarcual, which was determined by counting sets of spinal nerve foramina) and 104 tail vertebrae (n = 8 hatchlings counted; *Supplementary file 2*). These counts are consistent with the hypothesis that tail somites give rise to more (though not consistently double the number of) vertebral units compared with trunk somites. It should be noted, however, that there is difficulty in accurately counting terminal vertebrae in skates: the tail tapers to a fine point, with terminal vertebrae becoming extremely small and difficult to differentiate in skeletal preparations, and it is also likely that additional non-mineralized vertebral elements are present in the tip of the tail that we were not able to distinguish by this method. We therefore speculate that a

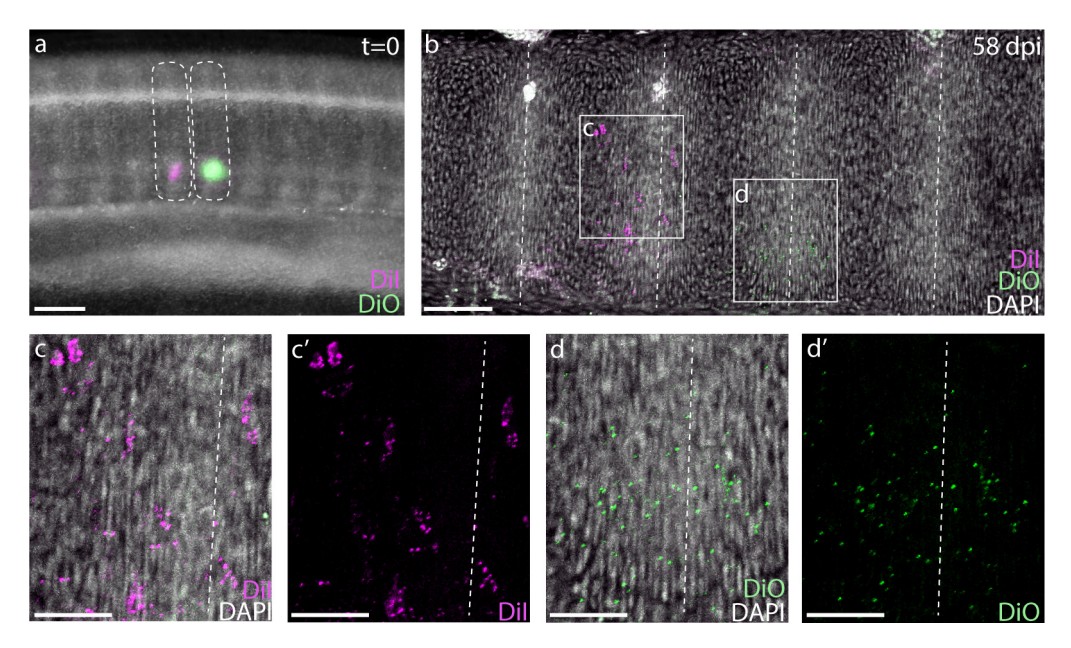

**Figure 3.** Skates undergo tetrapod-like resegmentation. (**a**) Adjacent CM-DiI- and SpDiOC$_{18}$-labeled trunk somites in a S24 skate embryo. (**b**) Sagittal section of the trunk of an embryo 58 days post-injection, showing CM-DiI- and SpDiOC$_{18}$-labeled cells in the posterior and anterior halves of adjacent vertebrae, consistent with resegmentation. (**c, c'**) CM-DiI-labeled cells and (**d, d'**) SpDiOC$_{18}$-labeled cells span two vertebrae (each shown with and without DAPI counterstain). Dashed lines indicate somite boundaries in (**a**) and vertebral boundaries in (**b-d**). Scale bars for (**a**) and (**b**) 100 μm; scale bars for (**c-d'**) 50 μm.

pattern of duplication of somite derivatives observed in the anterior tail may not persist along the full length of the tail, but rather breaks down near the tip.

To test whether a single somite gives rise to more than two vertebral halves in the presumptive diplospondylous vertebrae of the skate tail – and to assess whether the process of resegmentation occurs in the tail as in the trunk – we repeated the above somite labeling experiments in adjacent somites posterior to the cloaca (*Figure 4a*). Analysis of labeled embryos 10 weeks post-injection revealed CM-DiI and SpDiOC$_{18}$ each in the cartilage of three vertebrae: the posterior portion of one, the entire next successive vertebra, and the anterior portion of the third (n = 19/23, *Figure 4f–h*). Each tail somite therefore contributes to four vertebral halves spread over three vertebrae, fully double that of each trunk somite, and again consistent with a tetrapod-like model of somite resegmentation.

Because the reduction in spinal nerve foramina per vertebral segment coincides with the first caudal vertebra (*Figure 1b*), we hypothesized that the transition between the monospondylous vertebrae of the trunk and the diplospondylous vertebrae of the tail occurred at the cloaca. To test the location of this transition, we injected one of the posterior-most somites just dorsal to the cloacal opening with CM-DiI (*Figure 5a*). At 8–12 weeks post-injection, most cloacal-region somite injections resulted in the trunk-like resegmentation pattern of two vertebral halves derived from one somite (*Figure 5b*;c, n = 8/9). In each case, CM-DiI-labeled chondrocytes were recovered in vertebrae within several segments of the monospondylous-diplospondylous transition, suggesting that this boundary occurs just posterior to the cloaca.

## Discussion

Here we report the presence of strict somite resegmentation in a cartilaginous fish. Our two-color fate mapping experiments show clearly that a single skate trunk somite gives rise to adjacent halves of neighboring vertebral centra. When combined with fate mapping data from axolotl and chick, our findings indicate that vertebrae were formed through somite resegmentation in the last common

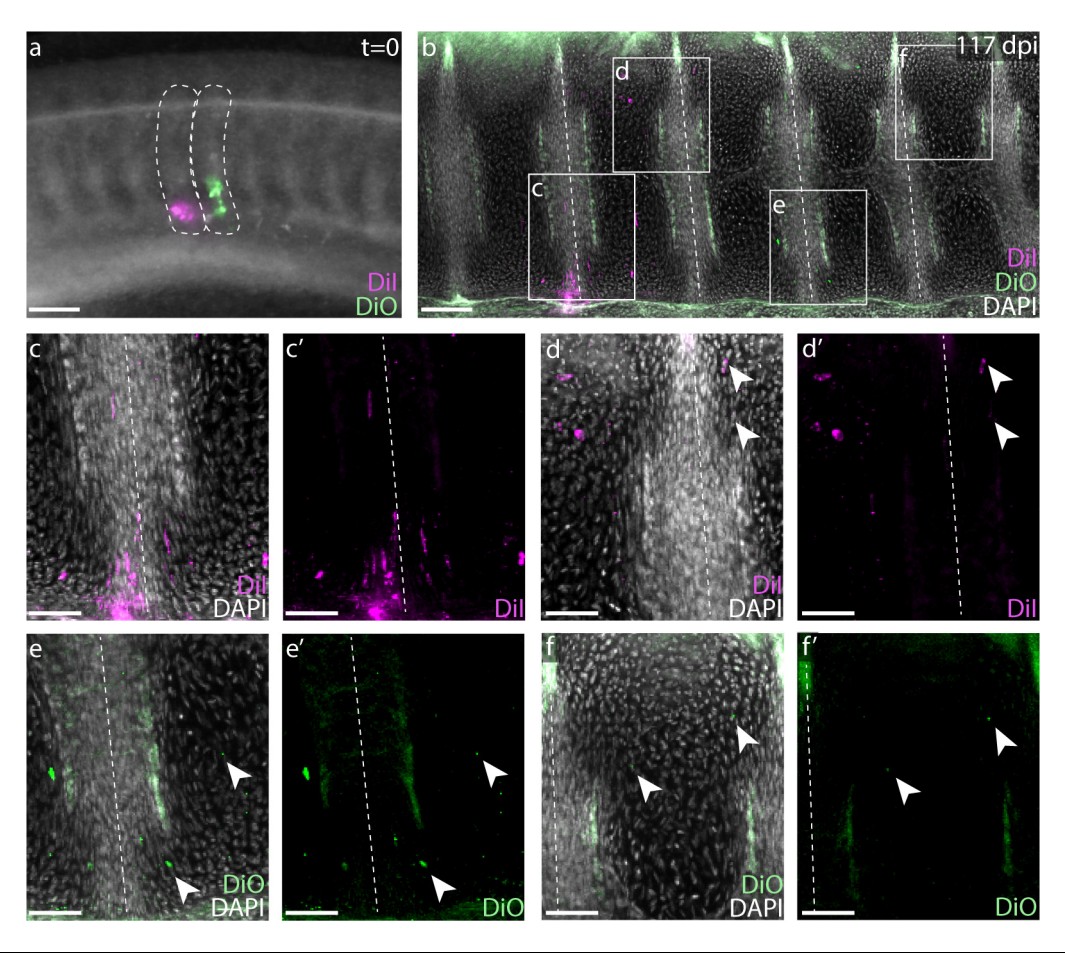

**Figure 4.** Skate tail vertebrae are duplicated and resegmented. (a) Adjacent CM-DiI and SpDiOC$_{18}$ injections in tail somites of a S24 little skate embryo. (b) Sagittal section of the tail of an embryo 117 days post-injection showing CM-DiI- and SpDiOC$_{18}$-labeled cells in three vertebrae each. Magnified views reveal (c-d) CM-DiI- and (e-f) SpDiOC$_{18}$-labeled cells spanning vertebral boundaries (images shown with and without DAPI counterstain). Dashed lines indicate somite boundaries in (a) and vertebral boundaries in (b–f). Arrowheads indicate dye-labeled cells. Scale bars for (a and b) 100 μm; scale bars for (c-f') 50 μm.

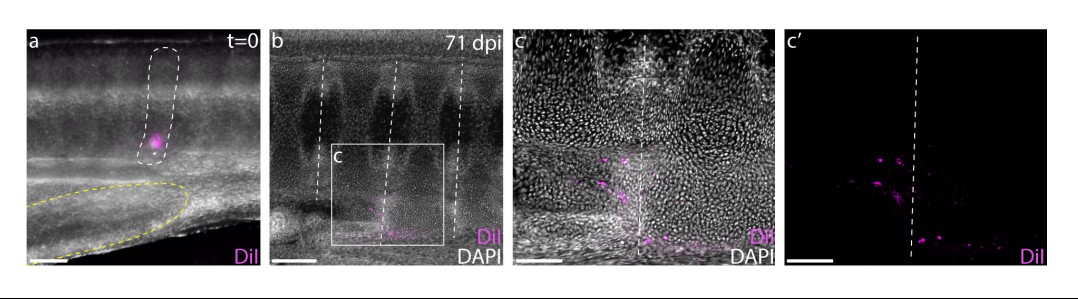

**Figure 5.** The cloaca precedes the monospondylous to diplospondylous transition. (a) CM-DiI labeling of a S24 skate somite (marked with white dashed line) just dorsal to the posterior margin of the cloaca (denoted with yellow dashed line). (b) Sagittal section of an embryo 71 days post-injection, showing two adjacent vertebrae labeled with CM-DiI. (c) Magnified view of two vertebrae showing CM-DiI labeled cells with and (c') without DAPI counterstain. Dashed lines in (b-c) indicate vertebral boundaries. Scale bars for (a) and (c) 100 μm; scale bars for (b) 200 μm.

ancestor of jawed vertebrates, and that the intermixing of somite cells throughout vertebral bodies seen in zebrafish represents a departure from that ancestral pattern (*Figure 6a*). We also show that resegmentation is maintained along the AP axis in skate, despite differences in the number of somite derivatives in the trunk and tail. In the trunk, each somite gives rise to two vertebral halves spanning a single vertebral boundary (*Figure 6b*), and this transitions posterior to the cloaca (*Figure 6c*) to tail somites giving rise to four vertebral halves spanning two vertebral boundaries (i.e. a single tail somite gives rise to a half vertebra, a full vertebra, and a subsequent half vertebra – *Figure 6d*). Unlike the ancestral pattern of resegmentation in the skate trunk, the modification of this arrangement to give rise to duplicated vertebrae in the tail is unique to elasmobranchs and, to our knowledge, previously unreported.

Establishment of somite polarity is of central importance to the development of the peripheral nervous system and to resegmentation and axial skeletal patterning. Vertebrate spinal nerve axons traverse exclusively through rostral half-somites upon exit from the central nervous system, with contact-mediated repulsion preventing axon migration through caudal half-somites (*Keynes et al., 1997*; *Rickmann et al., 1985*; *Steketee and Tosney, 1999*; reviewed in *Keynes, 2018*). This repulsive interaction is mediated, in part, by membrane-bound Eph family ligand-receptor interactions between caudal-half sclerotome and migrating neural crest cells/axons (*Araujo and Nieto, 1997*;

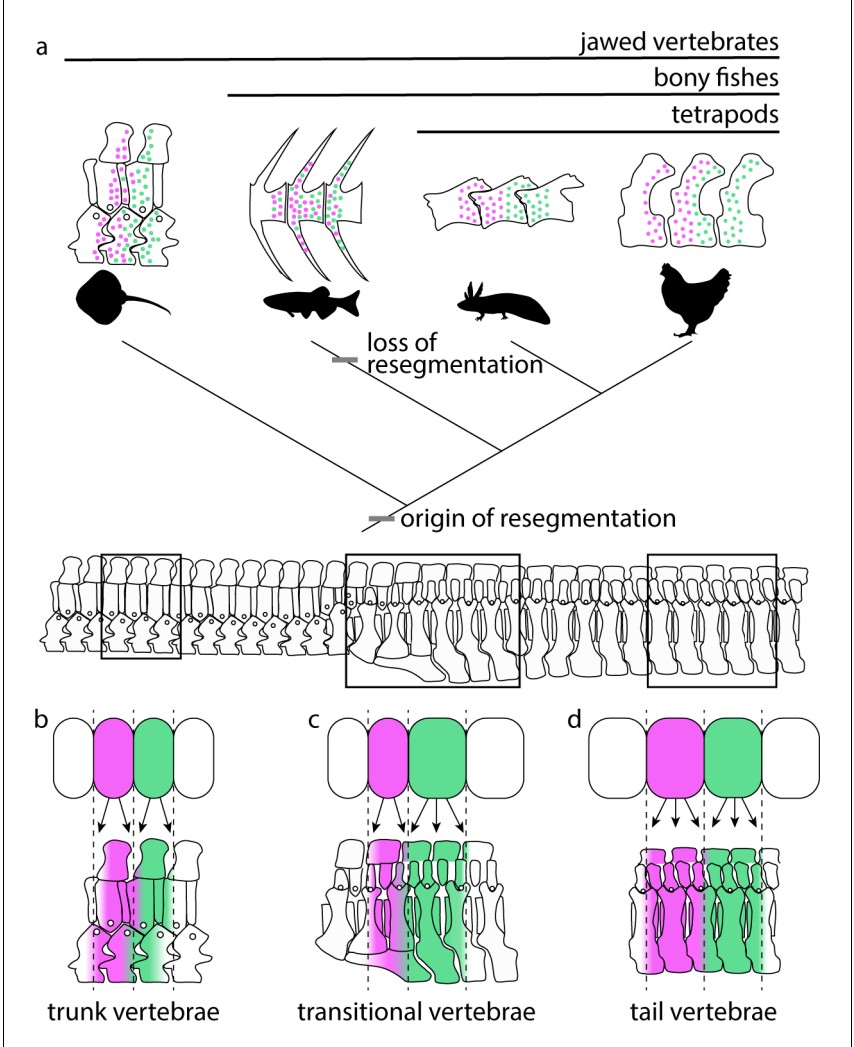

**Figure 6.** Resegmentation is ancestral for jawed vertebrates. (a) Schematic showing the distribution of strict resegmentation on a simplified vertebrate evolutionary tree, and the contribution of two adjacent somites to (b) trunk vertebrae; (c) vertebrae at the trunk-to-tail transition; and (d) tail vertebrae in the skate.

*Krull et al., 1997*; *Nieto et al., 1992*; *Wang and Anderson, 1997*), and by a cell surface glycoprotein that is expressed exclusively in caudal-half sclerotome (*Keynes, 2018*; *Rickmann et al., 1985*). It is likely that contact-mediated repulsion also underlies the recombination of sclerotome during resegmentation and vertebral development. Quail-chick chimera experiments have shown that only sclerotome derived from neighboring somite halves will intermingle following recombination – for example donor cells derived from rostral half-somites will intermingle with neighboring host sclerotome only when grafted next to another rostral half-somite, but will obey a tight boundary with no cell mixing when grafted next to a caudal half-somite (*Stern and Keynes, 1987*). This tendency for like cells to form discrete compartments within the somite likely underlies the subsequent contribution of rostral somite cells to caudal vertebrae, and vice versa, upon resegmentation. Interestingly, if this strict compartmentalization of somite halves is shared with skate, our finding of resegmentation in the diplospondylous tail region would suggest that rostral and caudal somite halves are not committed to caudal and rostral vertebral fates, respectively, but rather have the capacity to differentiate into either (as we would predict that the rostral half of a single somite would contribute to the caudal half of one vertebra and the rostral half of a second, neighboring vertebra, while the caudal half-somite would contribute to the caudal half of the second vertebra and the rostral half of a third). However, more specific lineage tracing experiments mapping the fates of half-somites are needed to test this.

Extensive molecular characterization of developing somites in bony vertebrates has revealed gene expression features corresponding with the distinct properties of rostral and caudal half-somites (*Hughes et al., 2009*; reviewed by *Kelly Kuan et al., 2004*). These features include polarized expression of notch signaling components (e.g. *Dll3* and *Dll1* in rostral- and caudal-half somites, respectively – *Bettenhausen et al., 1995*; *Dunwoodie et al., 1997*), FGF and BMP signaling components (e.g. FGFR1 in rostral-half somites and BMP5 and FGF3 and caudal-half somites – *Hughes et al., 2009*; *Mahmood et al., 1995*; *Yamaguchi et al., 1992*) and developmental transcription factors (e.g. *Id3* and *Tbx18* in rostral-half somites and *Twist*, *Meox-1* and *−2*, *Uncx4.1*, *Pax-1* and *−9* and *Sox9* in caudal-half somites – *Begemann et al., 2002*; *Bussen et al., 2004*; *Candia et al., 1992*; *Ellmeier and Weith, 1995*; *Füchtbauer, 1995*; *Haenig and Kispert, 2004*; *Hughes et al., 2009*; *Kraus et al., 2001*; *Mansouri et al., 2000*; *Müller et al., 1996*; *Schrägle et al., 2004*), and give rise to a transcriptional roadmap for the distinct rostral and caudal compartments of the mature somite. We have demonstrated conservation of polarized expression of *Tbx18* and *Uncx4.1* in the rostral- and caudal-half somites of cartilaginous and bony fishes, indicating that these features are likely ancestral for jawed vertebrates and a precondition for resegmentation. Interestingly, *Tbx15/18/22* and *Uncx4.1* are expressed in the rostral- and caudal-half somites, respectively, in the invertebrate chordate amphioxus, in a pattern similar pattern to that of vertebrates (*Beaster-Jones et al., 2008*). This indicates that the molecular basis of somite AP polarity may have arisen along the chordate stem, prior to the evolutionary origin of vertebrae. *Tbx15/18* is expressed in anterior somite halves in lampreys (*Freitas et al., 2006*), which also possesses a vertebral skeleton in the form of small, rudimentary neural arches. Lineage tracing experiments in lamprey to track the contribution of single somites to the neural arch cartilages could help to further resolve the evolutionary origin of resegmentation within vertebrates.

Historically, resegmentation was hypothesized to have originated through a functional need for the staggered positioning of myotomes and vertebrae in aquatic vertebrates. The attachment of muscle fibers from one myotome across a vertebral joint to two vertebral centra was suggested to facilitate lateral bending and therefore axial locomotion (von Ebner, as cited by *Fleming et al., 2015*). However, in fishes, the relative positioning of arches to centra, and of centra within myomeres, can vary substantially between species, and along the anteroposterior axis (*Schaeffer, 1967*; *Lauder, 1980*). Furthermore, myosepta in jawed vertebrates are morphologically highly complex, with a W-shape and six tendons that attach laterally to the skin and medially to the vertebrae and median septum, providing connections across up to three individual vertebrae (*Gemballa et al., 2003*; *Gemballa and Röder, 2004*). This anatomical complexity, in turn, renders it difficult to confidently infer spatial relationships between myomeres and vertebral units from adult morphology. It has also been speculated that diplospondyly in the caudal region of elasmobranch cartilaginous fishes is biomechanically advantageous, as a means of increasing flexibility in the tail during swimming, and that with additional vertebrae or centra in each myotomal segment, the tail could achieve greater control over the pattern of locomotion (*Lauder, 1980*; *Ridewood, 1899*; *Schaeffer, 1967*).

However, these hypotheses have not been tested using biomechanical models. Thus, while the phylogenetic distribution and functional advantages of staggered relationships between sclerotome and myotome derivatives – and of variation in vertebral unit-to-body segment ratios – remain unclear, our demonstration that resegmentation is an ancestral feature of the jawed vertebrate axial skeleton provides mechanistic insight into the evolution of axial skeletal segmentation, and developmental context for further comparative anatomical and biomechanical studies of the vertebrate axial column.

## Materials and methods

### Animal collection and husbandry

All skate embryos were obtained from captive brood stock at the Marine Biological Laboratory in Woods Hole, Massachusetts, USA and all experimental work was conducted in accordance with approved IACUC protocols. Embryos were reared in flow-through seawater tables at 10–12°C for approximately four weeks prior to experimentation at stage (S) 24. Early skate embryos were staged according to *Ballard et al. (1993)* and late-stage embryos were staged following *Maxwell et al. (2008)*.

### µCT scanning

Portions of the trunk and tail of a S34 skate embryo were stained with iodine potassium iodide (IKI) according to *Metscher (2009)*, and scanned using a GE v|tome|x µCT scanner at the University of Chicago. The trunk was scanned at 80 kV and 70 uA, with an exposure time of four seconds and a voxel size of 3.763 µm. The tail was scanned at 100 kV and 100 uA with a two second exposure and a voxel size of 3.075 µm. CT slices were processed and segmented in Avizo (ThermoFisher Scientific - FEI). Tiff stacks for each scan are available on the Dryad digital repository, doi:10.5061/dryad.b2rbnzs8s.

### Somite and vertebral counts

Vertebrae in hatchling skates were visualized by skeletal preparation, as described in *Gillis et al. (2009)* with an added overnight incubation in Alizarin red solution (1 mg/mL Alizarin red in 1% KOH) and overnight trypsin (1% w/v in water) digestion prior to KOH clearing. Somites in S25 skate embryos (n = 17) and vertebrae in cleared and stained skate hatchlings (n = 8) were imaged in numerous focal planes on a Leica M165 FC stereoscope. Image stacks were then merged in Helicon Focus Pro and tiled to form high resolution images. Somites and vertebrae were counted in Adobe Photoshop 2018 using a layered overlay and dots to mark individual elements.

### Histology and mRNA in situ hybridization

*L. erinacea* embryos were embedded in paraffin wax and sectioned at 8 µm thickness for mRNA in situ hybridization as described in *O'Neill et al. (2007)*. Chromogenic mRNA in situ hybridization experiments for *Uncx4.1* (GenBank accession number MN478366) and *Tbx18* (GenBank accession number MN478367) were performed on sections as described in *O'Neill et al. (2007)*, with modifications according to *Gillis et al. (2012)*. Probes, buffers, and hairpins for third generation in situ hybridization chain reaction (HCR) experiments were purchased from Molecular Instruments (Los Angeles, California, USA). Experiments were performed on paraffin sections according to the protocol of *Choi et al. (2018)*, with the following modifications: Following proteinase K treatment and rinsing, slides were pre-hybridized for 30 min at 37°C, and then hybridized overnight at 37°C with 0.8 µL of 1 µM probe stock/100 µL of hybridization solution. Following post-hybridization washes and pre-amplification steps, slides were incubated in amplification solution containing 4 µL of each hairpin stock/100 µL of amplification buffer.

### Fate mapping experiments

Two-color somite fate mapping experiments were performed as described in *Criswell et al. (2017b)* and *Ward et al. (2017)*. S24 skate embryos were removed from their egg cases to a petri dish and anesthetized in tricaine (MS-222 1 mg/L in seawater). Adjacent somites were injected with the red-fluorescent lipophilic dye CM-DiI and the green-fluorescent lipophilic dye SpDiOC$_{18}$ (ThermoFisher).

Concentrated stocks of CM-DiI (5 µg/µL in absolute ethanol) and SpDiOC$_{18}$ (2.23 µg/µL in dimethyl-formamide) were diluted 1:10 in 0.3 molar sucrose for injection. After injection embryos were returned to their egg cases and maintained in a flow-through seawater table at 15˚C for 8–12 weeks post-injection.

Injected skate embryos were euthanized using an overdose of tricaine (1 g/L in seawater) and fixed in 4% paraformaldehyde overnight at 4˚C. Embryos were then rinsed 3 × 5 min in phosphate-buffered saline (PBS), embedded in 15% gelatin in PBS and post-fixed in 4% paraformaldehyde in PBS for 4 nights at 4˚C before vibratome sectioning. A Leica VT1000S vibratome was used to cut 100 µm sections of tissue in sagittal plane, which were then DAPI-stained (1 µg/mL), coverslipped with Fluoromount-G (Southern Biotech) and imaged on an Olympus FV3000 (trunk and transitional vertebrae) or Leica Sp5 (tail vertebrae) confocal microscope.

# Acknowledgements

We thank Dr Victoria Sleight, Christine Hirschberger, and Jenaid Rees, along with the Cambridge Evolution and Development community for valuable feedback. We are grateful to Dr. Richard Schneider, Prof. David Sherwood, and the MBL Embryology Course for provision of lab space, Louise Bertrand and Leica Microsystems for microscopy support, and the staff of the Marine Resources Center at the MBL for aid in animal husbandry. This project benefited from use of the Imaging Facility, Department of Zoology, supported by a Sir Isaac Newton Trust Research Grant (Ref 18.07ii (c)).

# Additional information

## Funding

| Funder | Grant reference number | Author |
| --- | --- | --- |
| Royal Society | NF160762 | Katharine E Criswell |
| Royal Society | UF130182 | J. Andrew Gillis |
| Marine Biological Laboratory | | Katharine E. Criswell |

The funders had no role in study design, data collection and interpretation, or the decision to submit the work for publication.

## Author contributions

Katharine E Criswell, Conceptualization, Formal analysis, Funding acquisition, Investigation; J Andrew Gillis, Conceptualization, Funding acquisition

## Author ORCIDs

Katharine E Criswell https://orcid.org/0000-0002-4004-0192
J Andrew Gillis https://orcid.org/0000-0003-2062-3777

## Ethics

Animal experimentation: All experimental work was conducted at the Marine Biological Laboratory in Woods Hole, Massachusetts, USA, in accordance with approved institutional animal care and use (IACUC) protocols (#17-31 and #18-32). All embryological manipulations and euthanasia were performed with use of the anaesthetic Ethyl 3-aminobenzoate methanesulfonate (MS-222 or tricaine) and all efforts were made to minimise suffering.

## Decision letter and Author response

Decision letter https://doi.org/10.7554/eLife.51696.sa1
Author response https://doi.org/10.7554/eLife.51696.sa2

## Additional files

### Supplementary files
- Supplementary file 1. Table of counts of trunk and tail somites in S25 skate embryos.
- Supplementary file 2. Table of counts of trunk and tail vertebrae in hatchling skates.
- Transparent reporting form

### Data availability

Sequencing data have been deposited in GenBank (*Uncx4.1* accession number MN478366 and *Tbx18* accession number MN478367). CT scan data, including scan parameters and TIFF stacks, have been deposited in the Dryad Digital Repository, https://doi.org/10.5061/dryad.b2rbnzs8s.

The following datasets were generated:

| Author(s) | Year | Dataset title | Dataset URL | Database and Identifier |
|---|---|---|---|---|
| Criswell KE, Gillis JA | 2020 | Nucleotide sequences for Leucoraja erinacea Uncx4.1 | https://www.ncbi.nlm.nih.gov/nuccore/MN478366 | NCBI GenBank, MN478366 |
| Criswell KE, Gillis JA | 2020 | Nucleotide sequences for Leucoraja erinacea Tbx18 | https://www.ncbi.nlm.nih.gov/nuccore/MN478367 | NCBI GenBank, MN478367 |
| Criswell KE, Gillis JA | 2019 | Resegmentation is an ancestral feature of the gnathostome vertebral skeleton | http://doi.org/10.5061/dryad.b2rbnzs8s | Dryad Digital Repository, 10.5061/dryad.b2rbnzs8s |

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
