## [Decision Letter]

**Acceptance summary:**

This elegant study focuses on elucidating resegmentation-based vertebral patterning in vertebrate evolution. The novel findings are consistent with resegmentation, and as such provide new and exciting insight into this evolutionary process.

**Decision letter after peer review:**

Thank you for submitting your article "Resegmentation is an ancestral feature of the gnathostome vertebral skeleton" for consideration by *eLife*. Your article has been reviewed by two peer reviewers, and the evaluation has been overseen by a Reviewing Editor and Clifford Rosen as the Senior Editor. The following individual involved in review of your submission has agreed to reveal their identity: Roger Keynes (Reviewer #1).

The reviewers have discussed the reviews with one another and the Reviewing Editor has drafted this decision to help you prepare a revised submission.

Essential revisions:

Please include some discussion of whether there are other known or hypothesized A-P patterning genes regulating each half-somite, as assessed by in situ hybridisation for Tbx18/Uncx4.1 expression.

And please discuss whether there is evidence for skate myotome derivatives that straddle the intervertebral joints, and include discussion of possible advantages/disadvantages of diplospondyly.

---

## [Author Response]

Essential revisions:Please include some discussion of whether there are other known or hypothesized A-P patterning genes regulating each half-somite, as assessed by in situ hybridisation for Tbx18/Uncx4.1 expression.

Many other AP patterning genes have been demonstrated to regulate somite polarity in bony vertebrates. We have added to our discussion of these in two places: First, we have added text on the molecular basis of somite AP polarity with respect to axonal migration and patterning of the peripheral nervous system; and secondly, we have expanded our discussion of additional signaling pathways and transcription factors that establish somite AP polarity during development (Discussion).

And please discuss whether there is evidence for skate myotome derivatives that straddle the intervertebral joints, and include discussion of possible advantages/disadvantages of diplospondyly.

While it has been suggested that a functional consequence of axial resegmentation is the offset between myotome and sclerotome derivatives (such that myotome derivatives would straddle intervertebral joints), there is actually little evidence for this in amniotes (as discussed by Fleming et al., 2015). Additionally, we examined our existing collection of histological sections from skate embryos, but we were unable to confidently draw any conclusions regarding spatial relationships between myotome and sclerotome derivatives originating from the same or neighboring somites. Differentiated gnathostome myomeres are very complex in morphology, and we believe that the relationship between these structures and sclerotomal derivatives is complicated and difficult to infer from adult morphology alone. We suspect that clearly understanding this relationship would require a much more extensive study, for example using simultaneous lineage tracing of myotome and sclerotome, with fixation and analysis at multiple intermediate developmental time points (i.e. before the final state of differentiation is reached).

However, in light of the reviewer’s comment, we have substantially revised the final paragraph of our Discussion to address this, along with an additional discussion of the hypothesized functional advantage of resegmentation and diplospondyly.